# Adversarial Semantic Contour for Object Detection

**Yichi Zhang** [1]   **Zijian Zhu** [2]   **Xiao Yang** [1]   **Jun Zhu** [1] [*]

## Abstract

Modern object detectors are vulnerable to adversarial examples, which brings potential risks to numerous applications, e.g., self-driving car. Among attacks regularized by $\ell_p$ norm, $\ell_0$-attack aims to modify as few pixels as possible. Nevertheless, the problem is nontrivial since it generally requires to optimize the shape along with the texture simultaneously, which is an NP-hard problem. To address this issue, we propose a novel method of Adversarial Semantic Contour (ASC) guided by object contour as prior. With this prior, we reduce the searching space to accelerate the $\ell_0$ optimization, and also introduce more semantic information which should affect the detectors more. Based on the contour, we optimize the selection of modified pixels via sampling and their colors with gradient descent alternately. Extensive experiments demonstrate that our proposed ASC outperforms the most commonly manually designed patterns (e.g., square patches and grids) on task of disappearing. By modifying no more than 5% and 3.5% of the object area respectively, our proposed ASC can successfully mislead the mainstream object detectors including the SSD512, Yolov4, Mask RCNN, Faster RCNN, etc.

## 1. Introduction

Deep neural networks (DNNs) have demonstrated great power in object detection. Modern detectors can mainly be classified into two types, one-stage detectors (e.g., Yolo(Redmon et al., 2016; Redmon & Farhadi, 2016; 2018; Bochkovskiy et al., 2020)) and two-stage detectors (e.g., Faster R-CNN(Ren et al., 2016)). In particular, one-stage detectors are designed as a single neural network which

---

[1] Department of Computer Science and Technology, Institute for AI, THBI Lab, Tsinghua University, Beijing 100084, China [2] Institute of Image Processing and Network Engineering, Shanghai Jiao Tong University, Shanghai 200240, China. Correspondence to: Jun Zhu <dcszj@mail.tsinghua.edu.cn>.

*Accepted by the ICML 2021 workshop on A Blessing in Disguise: The Prospects and Perils of Adversarial Machine Learning.* Copyright 2021 by the author(s).

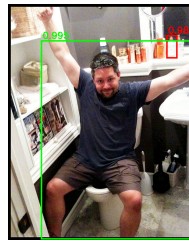 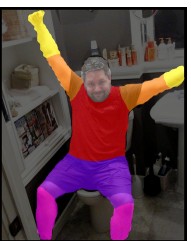 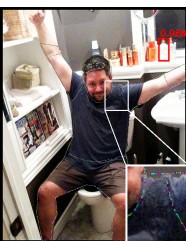

(a) Detection with Clean Image  (b) Part Segmentation by CDCL (Lin et al., 2020)  (c) Invisible to detector with ASC Noises

*Figure 1.* To attack object detectors, we focus on the object contour area, which is the general shape and outline of an object. To acquire the object contour, we adopt part segmentation method (CDCL (Lin et al., 2020)) to produce object semantic contour. By optimizing the pixel selection around the object contour and the colors, we can have the optimized Adversarial Semantic Contour (ASC) to successfully cloak the person from Faster RCNN (Ren et al., 2016) as the figure shows. To show the details of our method, we zoom in the part inside the white rectangle.

predicts bounding boxes and classification directly from the image. The prediction of two-stage detectors consists of region proposals and classification. Compared with one-stage detectors, two-stage detectors are generally more accurate and robust but slower with prediction as reported in (Redmon & Farhadi, 2018). For Faster R-CNN, with redundant candidate proposals from Region Proposal Network (RPN), the prediction can still be accurate even though some of the proposals are disturbed making the attack difficult.

However, modern detectors can be affected by carefully designed adversarial perturbations, bringing potential safety threats to real-world applications. Mainstream methods can be categorized according to the norm used for regularization. The earliest methods which reveal the weakness of DNNs (e.g., PGD (Madry et al., 2019)) mostly adopt $\ell_\infty$ norm, bounding the perturbations to be imperceptible to human. Besides, adversarial noises can be regularized by $\ell_0$ norm, modifying a limited number of pixels, while the perturbation may not be bounded. The main purpose of this paper is to improve the performance of $\ell_0$ attack on object detectors, because $\ell_0$ attack modifies few pixels, which is more meaningful for object detection. In general, $\ell_0$ optimization is a typical Non-deterministic Polynomial hard problem (NP-hard) which makes the optimization non-differential

and time-consuming. Therefore, most of previous works on $\ell_0$ attack modify pixels that meet certain patterns which are designed in advance. Nevertheless, these manually designed patterns cannot guarantee successful attack depending on various factors like the implementation of models, especially for two-stage detectors.

To address these challenges, we propose a novel method of Adversarial Semantic Contour (ASC), which improves the $\ell_0$ attack performance on object detectors. Different from the previous works, we present a generic solution which can optimize the attack area with enough semantics. To make $\ell_0$ optimization efficient, we introduce the object contour as prior to reduce the searching space, which guides the further optimization. The reason why we take object contour as prior is that in "What is an object?" (Alexe et al., 2010), the authors proposed three characteristics of an object, including having closed boundaries and different appearance against the background, which means object contour carries semantic information of an object. Technically, based on the prior contour, we optimize the selection of modified pixels via sampling and their colors with gradient descent alternately. When it converges, we get the Adversarial Semantic Contour and achieve successful attack.

We conduct comprehensive experiments to evaluate the performance of our algorithm. Object localization is what distinguishes object detectors from classifiers. Therefore we design the task of disappearing, which aims to cloak the object completely from the detectors. We select square patch as a comparison method, which is one of the most commonly used methods in $\ell_0$ attack on object detection, e.g., AdvPatch (Thys et al., 2019). Also, we take grid-like pattern from DPAttack (Wu et al., 2020), which got the second place in the CIKM2020 AnalytiCup "Adversarial Challenge on Object Detection". From the result, by setting the threshold of $\ell_0$ norm at 5% and 3.5% of object area respectively, our method, ASC, outperforms these methods on four main-stream detectors, including one-stage detectors like SSD512 (Liu et al., 2016) and Yolov4 (Bochkovskiy et al., 2020), and two-stage detectors like Mask RCNN (He et al., 2018) and Faster RCNN (Ren et al., 2016).

In summary, we make the technical contributions as:

- Different from the existing methods that use patterns with fixed shapes, we propose a new $\ell_0$ adversarial attack method on object detectors which optimize the selection and texture of modified pixels jointly.

- In order to avoid the exhausting search over the whole image, we introduce the detected contour as prior to reduce the searching space. Guided by the prior contour, we optimize the pixel selection via sampling and the texture with gradient descent alternately, which improves the efficiency of optimization and increases the

chance of successful attack.

## 2. Methodology

Our goal is to develop a pattern to carry out more effective $\ell_0$ attack on object detectors, which modifies as fewer pixels as possible while guarantees successful attack. Here, we present our formulation of $\ell_0$ attack and our contour-based method.

### 2.1. Problem Formulation

$\ell_0$ **Attack on Object Detection.** In general, to attack the object detectors with $\ell_0$ regularization, we need to decide the pixels to be disturbed, which should be as few as possible, and their adversarial textures which can corrupt the prediction of the detector most. We consider a formulation of finding the best adversarial noises for each image in a general form as

$$(M^*, T^*) = \underset{P=(M,T)}{\arg\max} J\Big(f_\theta\big(x \oplus P\big), y\Big) - \alpha \ell_0(M), \quad (1)$$

where $x$ is the original image, $y$ is its corresponding ground-truth label, $f_\theta$ denotes the pretrained detector with $\theta$ representing the structure and weights of the model, $J(f_\theta(x), y)$ denotes the objective function which describes the similarity between output and label, $M$ is a 0-1 mask in which each value represents whether a pixel is selected to be modified, $T$ is a matrix of the same size with $x$ which represents the colors for each pixel, and $x \oplus P = x \oplus (M, T)$ is a simplified form of $(1 - M) \cdot x + M \cdot T$ which is more accurate mathematically. Our goal is to maximize the objective function $J$ to corrupt the detector prediction.

However, among these attributes, the optimization of $M$ is an $\ell_0$ problem and cannot reach its optimal points by traditional gradient descent methods, which makes the optimization of these nondifferentiable parameters a Nondeterministic Polynomial hard problem (NP-hard). This will lead to low efficiency in our attack if we apply global search over the image with blindness. To release this problem, we review the problem with a Bayesian perspective and introduce a prior which may narrow the searching space and improve the efficiency of $\ell_0$ optimization, while still being able to find an approximate optimal solution.

### 2.2. Using Object Contour?

With reference to "What is an object?" (Alexe et al., 2010), we notice the proposed attributes of objects and evaluation metrics on object detection. Among them, the attributes, including closed boundaries and significance against background, are closely related to the boundary between object and background, while two of the metrics, Color Contrast and Edge Density, also take use of the object significance

and contours. Thus, we believe the contours carry enough object semantics and can be taken as the focus area. Technically, with the modern methods of part segmentation and contour detection, we take the detected contour map as the prior and denote it as $M_p$ which is a prior case for $M$ introduced in Sec.2.1. We don't adopt instance segmentation or edge detection because we want to acquire the exact and clear boundaries between different parts of the object. We will make use of the prior knowledge carried with the contour map to optimize the selection of pixels to be modified and their colors to attack the detector by turns.

For a 0-1 matrix $M$, we define a set of pixel coordinates whose corresponding values in $M$ are 1 as $\mathbb{M} = \{(x_1, y_1), (x_2, y_2), ..., (x_n, y_n)\}$. Clearly, our optimization restriction, $\ell_0(M)$, equals to $|\mathbb{M}|$, the size of $\mathbb{M}$. With the semantic information in the prior $M_p$, we believe the approximate optimal solution of pixel selection should be near the pixels in $\mathbb{M}_p$. By optimizing the colors of the selected pixels and observing the performance, we can evaluate the attack effectiveness of the pixel set $\mathbb{M}$. Thus, we may optimize the adversarial noises iteratively. Starting from the prior contour pixels $\mathbb{M}_0 = \mathbb{M}_p$, we may sample around these pixels, acquire new pixel sets, evaluate their effectiveness and decide whether to update $\mathbb{M}_k$ until we get an approximate optimal solution.

Based on the above analysis, our optimization of contour involves the contour map $M$ and the corresponding color $T$ separately and alternately. Assuming we have set a contour map $M$, we adopt a method similar to Projected Gradient Descent (PGD) (Madry et al., 2019) to optimize the color, which projects the updated color $T$ into the acceptable range for pixel values. With more details, we update the adversarial noises following the equation,

$$T_{i+1} = \text{Clip}_{[0,1]}\Big(T_i + \alpha \cdot \nabla_x J\big(f_\theta(x \oplus (M, T_i), y)\big)\Big), \quad (2)$$

where we only optimize $T$ with gradients of objective function $J$ for a given $M$ and $\alpha$ is a hyperparameter as step size. For a image with pixel values ranging from 0 to 1, we clip the modified pixel values within the acceptable range after every updating step. Thus, we can optimize the color $T$ iteratively with gradients to attack the object detectors.

Therefore, we formulate our problem into a prior-guided iterative optimization with object contour, which circumvents the low-efficient $\ell_0$ optimization and improve the chance of successful attack due to the object semantic information.

## 3. Experiments

We conduct a series of experiments on four different mainstream detectors to prove that adversarial attack based on Adversarial Semantic Contour can achieve more satisfying performance than the most popular existing patterns.

### 3.1. Experiment Settings

**Dataset.** We select 1000 images from Microsoft COCO2017 (Lin et al., 2015). In consideration of the previous studies and the convenience to acquire object contours, we only attack these 1000 objects categorized as "Person". We believe this setting is reasonable and with no loss of generality.

**Models.** We attack 4 models in total, including 2 two-stage detectors (Faster R-CNN (Ren et al., 2016) and Mask R-CNN (He et al., 2018)) and 2 one-stage detectors (SSD512 (Liu et al., 2016) and Yolov4 (Bochkovskiy et al., 2020)). Concretely, Yolov4 (Bochkovskiy et al., 2020) is a pytorch implementation[1], while the other three detectors are implemented by mmdetection[2] (Chen et al., 2019).

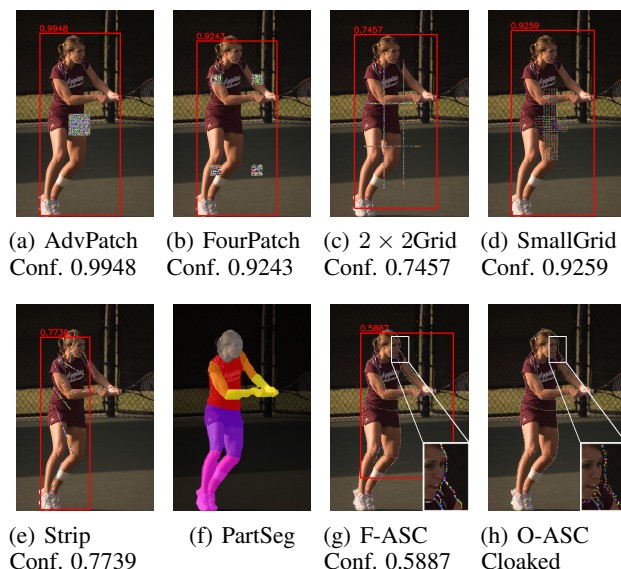

(a) AdvPatch Conf. 0.9948    (b) FourPatch Conf. 0.9243    (c) $2 \times 2$Grid Conf. 0.7457    (d) SmallGrid Conf. 0.9259

(e) Strip Conf. 0.7739    (f) PartSeg    (g) F-ASC Conf. 0.5887    (h) O-ASC Cloaked

*Figure 2.* Attack performance of 6 patterns. All comparison patterns fail to cloak the person from Faster RCNN (Ren et al., 2016). Though also failed, the fixed ASC (F-ASC) depresses the objectness confidence the most comparing to other patterns. After adjusting the contour shape with ground-truth label and sampling around the contour, we have the optimized ASC (O-ASC) which successfully makes the person invisible to the detector.

**Adversarial Patterns.** Our methods (ASC) is based on the semantic boundaries on objects. For prior contour map mentioned in Sec.2.2, we acquire it with CDCL (Lin et al., 2020). We demonstrate two cases, among which the "F-ASC" stands for the fixed case that directly uses contours generated with CDCL as target pixels and the "O-ASC" stands for the optimized case that consists of the iterative optimization introduced in Sec.2.2. What worths notice is that we only carry out the complete optimization on examples that fail to be attacked as mentioned with "F-ASC".

---

[1]https://github.com/Tianxiaomo/pytorch-YOLOv4
[2]https://github.com/open-mmlab/mmdetection

To verify the effectiveness of our contour-based attack, we select five patterns for comparative experiments. To be specific, we use "AdvPatch" and "FourPatch" for square patches referring to (Thys et al., 2019) and (Huang et al., 2020) and "2 × 2Grid" and "SmallGrid" for grid-like patterns referring to (Wu et al., 2020). Additionally, we use the pattern, "Strip", which is composed of the diameter lines of the part segmentation of the object. The intention of setting this pattern is to prove that the contour area is more effective, comparing to inner area of the segmentation. Since we stress the $\ell_0$ norm, we have the area cost budget bounded by 3.5% and 5% of the object area respectively and the area cost of the five comparison patterns is restricted to be no less than the contour pattern, which is equivalent to our optimization formulation in Sec.2. An example of forms and results of 6 patterns is given in Fig.3.1.

**Metric.** We define "detected" as the case where the detector can make a prediction which has the IoU above 0.5 with the ground-truth label and has the confidence more than 0.5. We use the Successful Detection Rate (SDR) as the metric to evaluate the performance in this task. For one image, if the detector gives a prediction which makes the target "detected", the image is counted as a successful detection.

### 3.2. Performance

We aim to make the objects invisible to the detectors. To some extents, this is an extreme case of targeted attack, where we try to maximize the possibility of the object being the background. We set the loss function in Eq.1 for an image as

$$J(f_\theta, y) = -\sum_i \log(p_i), \qquad (3)$$

where $p_i$ is the objectness score of the $i^{th}$ bounding box which has the IoU above 0.5 with the ground-truth label.

*Table 1.* Successful Detection Rate (%, ↓) in task of disappearing. On two-stage detectors, the fixed case of ASC outperforms other patterns with at least a gap of 3% while performing above average on one-stage detectors. With further optimization, performance of our method increases and takes the lead on all detectors.

| Pattern | FRCN | | MRCN | | SSD512 | | Yolov4 | |
|---|---|---|---|---|---|---|---|---|
| $\ell_0$ budget(%) | 5.0 | 3.5 | 5.0 | 3.5 | 5.0 | 3.5 | 5.0 | 3.5 |
| Clean | 98.3 | | 98.9 | | 95.6 | | 91.4 | |
| AdvPatch | 70.6 | 86.8 | 74.8 | 89.6 | 2.0 | 15.6 | 2.6 | 23.8 |
| FourPatch | 55.0 | 78.6 | 58.1 | 81.2 | 3.5 | 34.8 | 3.9 | 29.6 |
| SmallGrid | 33.7 | 54.2 | 40.8 | 60.2 | 1.1 | 6.1 | 1.7 | 5.3 |
| 2 × 2Grid | 12.7 | 24.6 | 14.5 | 29.5 | 0.3 | 1.9 | 2.0 | 5.1 |
| Strip | 15.1 | 38.6 | 17.2 | 45.0 | 1.2 | 6.8 | 0.7 | 7.6 |
| F-ASC | 9.7 | 13.4 | 10.8 | 14.7 | 1.1 | 2.2 | 1.2 | 4.6 |
| O-ASC | **2.0** | **6.6** | **2.7** | **7.4** | **0.1** | **0.9** | **0.0** | **1.4** |

Table 1 shows the the results of our experiments with different patterns. The two methods with square patches perform

the worst on the four models, especially with a large gap on two-stage detectors. The other two methods with grids have better performance comparing to the square patch methods and the performance of "2 × 2Grid" group improves more with SDR dropping down to 12.7% on Faster RCNN. The "Strip" group also performs better than average among the 5 comparison groups. As for our method, the fixed ASC outperforms all 5 comparison groups with at least SDR of 3% on two-stage detectors and have performance above average on one-stage detectors. After optimizing the examples which fail in the fixed case according to the shape optimization method mentioned in Sec.2.2, we see that the performance of ASC improves further. From the data with 5% $\ell_0$ constraint, our method leads on all models and the gaps between the minimum SDR of comparison groups and ours are widened, from 12.7% to 2.0% on Faster RCNN, from 14.5% to 2.7% on Mask RCNN, from 0.3% to 0.1% on SSD512 and from 0.7% to 0.0% on Yolov4. By restricting the $\ell_0$ budget from 5% to 3.5%, the gap between our method and other comparison methods becomes significantly greater, indicating with stricter $\ell_0$ bound, our method (ASC) can be more effective. Experimental result proves that using contour area as prior knowledge to attack object detectors has more effective performance than other fixed patterns which are designed manually, since it carries more object semantic information. Meanwhile, having smaller SDR than "Strip" group provides further evidence that object detectors can be more sensitive to object contours than object inner area.

## 4. Conclusion

Objects have clear and definite attributes, among which having closed boundaries is a significant one. Therefore, detectors which need to localize objects are highly likely to learn using contours. Thus, in this paper, we propose the Adversarial Semantic Contour (ASC) as a new method, which does joint optimization of pixel selection and texture, to carry out $\ell_0$ attack on object detectors. We introduce the contour area as the prior knowledge and optimize the adversarial pattern based on that, to avoid the challenge that $\ell_0$ optimization is NP-hard. With the guide of prior contour, we optimize the selection of pixels to be modified by sampling and their texture with gradient descent alternately. From the concrete experiments, we see that with limitation of $\ell_0$ norm, ASC outperforms other existing patterns (square patch and grid), especially with a large margin on two-stage detectors. This proves that optimizing both shape and color of the contour pattern which is full of semantic information is more effective in $\ell_0$ attack on object detectors. We believe that object detectors are more vulnerable and sensitive to attack around the contour area which is relatively easy to be realized in real-world scenarios and may worth further research.

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
