# OpenReview forum: "Adversarial Semantic Contour for Object Detection"
_ICML.cc/2021/Workshop/AML — ICML 2021 Workshop AML Poster_

### Official Review · Reviewer_3oyL · 2021-06-20
**Interesting main idea**

**Rating:** Accept
**Confidence:** 5

**Review:**

This paper proposed an l_0 regularized adversarial attack, Adversarial Semantic Contour (ASC) to deceive object detectors on the ‘Person’ class. The authors used a part segmentation algorithm to produce the semantic contour map as the initial set of the perturbated pixels. They iteratively updated the pixel set and pixel values to get the optimal adversarial perturbation under a certain l_0 constrain. This attack outperforms patch-based attack algorithms and other attacks with certain patterns.

Strengths:
1. This paper is well organized and well-supported by experiments.
2. This method is inspired by an interesting thought that the contours of objects carry enough object semantics and can be taken as the focus area. The authors proved the effectiveness of the contours experimentally by comparing several patch-based and other attacks with certain patterns.

Weaknesses:
1. The reason for using part segmentation instead of other algorithms like instance segmentation or directly use edge detection is not clarified.
2. The comparison of other attacks with certain patterns can be fairer, e.g. shuffle the contour lines between different bounding boxes or directly flip the contour, which maintains the dispersion of the pixels in the pattern.

---

### Decision · Program_Chairs · 2021-06-21

**Decision:**

Accept (Poster)

**Comment:**

This paper proposed l0 adversarial attack to fool object detectors. The authors can further address the reviewer's comments in the revision.